# Activity-based probe profiling of RNF12 E3 ubiquitin ligase function in Tonne-Kalscheuer syndrome

Francisco Bustos*, Sunil Mathur*, Carmen Espejo-Serrano, Rachel Toth, C James Hastie, Satpal Virdee, Greg M Findlay

**Ubiquitylation enzymes are involved in all aspects of eukaryotic biology and are frequently disrupted in disease. One example is the E3 ubiquitin ligase RNF12/RLIM, which is mutated in the developmental disorder Tønne-Kalscheuer syndrome (TOKAS). RNF12 TOKAS variants largely disrupt catalytic E3 ubiquitin ligase activity, which presents a pressing need to develop approaches to assess the impact of variants on RNF12 activity in patients. Here, we use photo-crosslinking activity-based probes (photoABPs) to monitor RNF12 RING E3 ubiquitin ligase activity in normal and pathogenic contexts. We demonstrate that photoABPs undergo UV-induced labelling of RNF12 that is consistent with its RING E3 ligase activity. Furthermore, photoABPs robustly report the impact of RNF12 TOKAS variants on E3 activity, including variants within the RING domain and distal non-RING regulatory elements. Finally, we show that this technology can be rapidly deployed in human pluripotent stem cells. In summary, photoABPs are versatile tools that can directly identify disruptions to RING E3 ubiquitin ligase activity in human disease, thereby providing new insight into pathogenic mechanisms.**

## Introduction

Protein ubiquitylation involves the ligation of the 76 amino acid protein ubiquitin to specific protein substrates via a three-step enzymatic cascade (Hershko et al, 2000). This reaction requires ATP consumption and E1 activating (E1), E2 conjugating (E2), and E3 ligase enzymes (E3s) (Pickart, 2001). Of these, E3s represent a key regulation point as they confer substrate specificity (Zheng & Shabek, 2017). There are two main classes of E3 with distinct mechanisms of action. Transthiolation E3s, including HECT (homologous to E6AP carboxyl terminus), RBR (RING-in-between-RING), and RCR (RING-Cys-Relay) type enzymes, catalyse the transfer of ubiquitin to substrate after covalent ligation of ubiquitin to a catalytic cysteine residue (Scheffner et al, 1995; Dove & Klevit, 2017; Pao et al, 2018). RING (Really interesting new gene) E3s use an adapter-like mechanism and catalyse direct transfer of ubiquitin from the E2 to the substrate (Budhidarmo et al, 2012; Metzger et al, 2014).

RING E3s are the largest class of ubiquitin ligases and have been implicated in diseases including cancer, degenerative pathologies and developmental disorders (Basar et al, 2021; Cockram et al, 2021; Dang et al, 2021; Schmidt et al, 2021). How RING E3 ligases are regulated and dysregulated in disease is poorly understood, partly due to the paucity of methods for measuring cellular RING E3 activity. An emerging disease area in which protein ubiquitylation plays a prominent role is intellectual disability (Salvador-Carulla et al, 2011), which comprises a series of neurodevelopmental disorders affecting adaptive and cognitive function of affected individuals. These highly debilitating conditions affect 1–3% of the world's population and have no cure or treatment (Ropers & Hamel, 2005; Picker & Walsh, 2013). A significant proportion of these disorders have an underlying genetic origin (Delaunoy et al, 2001; Hackett et al, 2010; Ellison et al, 2013; Gonçalves et al, 2014; Luco et al, 2016) with frequent disruption to genes encoding ubiquitylation components, underscoring its importance in this disease setting (Froyen et al, 2008; Geetha et al, 2014; Bramswig et al, 2017; Santiago-Sim et al, 2017; Tsurusaki et al, 2017; Zhang et al, 2017; Fountain et al, 2019; Geerts-Haages et al, 2020; Johnson et al, 2020; Suzuki et al, 2020; Beck et al, 2021; Li et al, 2021).

Tønne-Kalscheuer syndrome (TOKAS; MIM #300978) is one such intellectual disability disorder caused by pathogenic variants in the gene encoding the RING E3 RNF12/RLIM (Tønne et al, 2015; Hu et al, 2016; Bustos et al, 2021). TOKAS is an X-linked syndromic form of intellectual disability, which also presents congenital abnormalities such as facial dysmorphism, abnormalities of the hands, lungs and gonads, and diaphragmatic hernia in utero, which may result in perinatal death (Tønne et al, 2015; Hu et al, 2016). RNF12 is expressed in a tissue-specific manner with enrichment in the brain (Bustos et al, 2020) and is predominantly localised in the nucleus (Jiao et al, 2013). RNF12 performs key developmental functions including regulation of gene dosage compensation via imprinted X-chromosome inactivation (Gontan et al, 2012, 2018) and regulation of developmental gene expression programmes (Zhang et al, 2012; Bustos et al, 2018, 2020; Segarra-Fas et al, 2020 *Preprint*). RNF12 ubiquitylates several substrates, including REX1 (Gontan et al, 2012), HDAC2 (Krämer et al, 2003), CLIM1/2 (Ostendorff et al, 2002), TRF1 (Her & Chung, 2009), Stathmin (Chen et al, 2014; Li et al, 2015), ERα

---

Medical Research Council (MRC) Protein Phosphorylation and Ubiquitylation Unit, University of Dundee, Dundee, UK

Correspondence: g.m.findlay@dundee.ac.uk; s.s.virdee@dundee.ac.uk
*Francisco Bustos and Sunil Mathur contributed equally to this work.

---

(Johnsen et al, 2009), c-Myc (Gao et al, 2016), BRF1 (Wang et al, 2019), and SMAD7 (Zhang et al, 2012). REX1 is a substrate that may be relevant to TOKAS aetiology, as it is a transcriptional regulator that controls neurodevelopmental gene expression (Bustos et al, 2020).

Thus far, most identified RNF12 TOKAS variants are missense mutations in two functional regions; the RING domain, which confers E3 ubiquitin ligase activity, and a basic region implicated in substrate recognition (Ostendorff et al, 2002; Gontan et al, 2012; Tønne et al, 2015; Hu et al, 2016; Bustos et al, 2018, 2021; Frints et al, 2019). Interestingly, this region is also involved in allosteric regulation of catalysis (Bustos et al, 2018). TOKAS variants in either region impair RNF12 substrate ubiquitylation (Bustos et al, 2018; Frints et al, 2019), leading to disruption of developmental gene expression (Bustos et al, 2020; Segarra-Fas et al, 2020 Preprint).

A major challenge in clinical settings is determining the impact of disease-associated genetic variants on protein function, and therefore establishing likely pathogenic mechanisms. Activity-based probes (ABPs) emerged as powerful tools for studying components of the ubiquitin system, including E3s (Borodovsky et al, 2002; Mulder et al, 2016; Pao et al, 2016; Xu et al, 2019). By specifically labelling the active conformation, cellular E3 activity is directly assessed independently of substrate and upstream E1 and E2 enzymes (Henneberg & Schulman, 2021). An ABP for transthiolation E3 ubiquitin ligases has been exploited in the context of Parkinson's disease to dissect the determinants of activation for the RBR E3 Parkin and shows that disease variants disrupt transthiolation activity (Pao et al, 2016). We have also described photocrosslinking activity-based probes (photoABPs) for RING E3s (Mathur et al, 2020). These photoABPs consist of an engineered ubiquitin-loaded E2 carrying a strategically positioned photocrosslinking group that were shown to specifically label the active forms of the RING E3s RNF4 and Cbl (Mathur et al, 2020). However, the utility of RING-specific photoABPs in defining pathogenic variants in RING E3 ubiquitin ligases has not yet been explored.

Here, we use photoABP technology to determine the impact of human disease variants on catalytic activity of the RING E3 RNF12. We demonstrate that a photoABP robustly monitors RNF12 activity in vitro and in situ of cell extracts, in a manner that is consistent with its catalytic mechanism. Importantly, the photoABP reports disruptions to RNF12 E3 activity caused by pathogenic TOKAS variants, including those in cryptic allosteric regulatory regions. Finally, we demonstrate utility of the photoABP to monitor the impact of a pathogenetic variant in an RNF12 TOKAS knock-in cell line model, and to report RNF12 E3 activity in a human induced pluripotent stem cell (hiPSC) line. In summary, we demonstrate the application of photoABP technology to profile human disease variants in a RING E3 ubiquitin ligase, which could be expanded to provide unprecedented insight into the dysregulation of the ubiquitin system in human disease and used as a disease biomarker.

# Results

### Monitoring activity of the disease associated RNF12 E3 ubiquitin ligase using a RING-specific photocrosslinking activity-based probe (photoABP)

A major challenge in the ubiquitin field is robust and sensitive measurement of enzymatic activity in a cellular context. We previously engineered RING E3 photoABPs consisting of a ubiquitin-loaded E2 where a photocrosslinking group has been incorporated into the ubiquitin component, enabling UV-dependent covalent labelling of the RING E3 (Mathur et al, 2020). The E2-ubiquitin cysteine thioester bond is also chemically replaced by a stable lysine isopeptide. Labelling occurs specifically in the context of the catalytically productive ternary complex where a closed E2~Ub conformation is stabilised by a RING E3. Importantly, stabilisation can only be achieved when the RING E3 is active (Mathur et al, 2020) (Fig 1A). This has previously been shown to specifically report cellular Cbl RING E3 activity (Mathur et al, 2020). However, the utility of photoABPs to investigate disruption of RING E3 enzymatic activity by human disease variants has not been explored.

Variants in the RING E3 RNF12 cause the X-linked intellectual disability disorder Tønne-Kalscheuer syndrome (TOKAS) (Tønne et al, 2015; Hu et al, 2016; Frints et al, 2019). As RNF12 TOKAS variants largely disrupt catalytic activity (Bustos et al, 2018; Frints et al, 2019), we used RNF12 as an exemplar by which to determine whether photoABP technology can be used to interrogate the impact of human disease variants on RING E3 activity. We first sought to determine whether RNF12 E3 activity can be monitored using photoABP technology. Initially, we took recombinant full-length RNF12, which is catalytically active (Bustos et al, 2018), and incubated with photoABP-Bpa31 (hereinafter referred to as photoABP, [Mathur et al, 2020]) followed by UV photocrosslinking. PhotoABP is engineered from UBE2D3, which is a member of the major family of E2s that participate in RNF12-dependent ubiquitylation (Bustos et al, 2018). This led to retarded RNF12 electrophoretic mobility, consistent with the mass expected for photoABP labelling (Fig 1B). This occurred in a manner similar to a constitutively active RNF4 consisting of the protein fused to an additional RING domain, RNF4*, which served as a positive control (Plechanovová et al, 2011; Mathur et al, 2020), albeit RNF12 is labelled to a lesser extent (Fig 1B). We determined the sensitivity and dynamic range of RNF12 photoABP labelling and established that optimal concentrations for immunoblot visualisation are 1 $\mu$M RNF12 and 20 $\mu$M photoABP (Fig 1C and D). Therefore, like other RING E3 ubiquitin ligases, RNF12 can be sensitively and specifically labelled by photoABP in an activity-dependent manner. It should be noted that a minor fraction of RNF12 is probe-labelled in these experiments. This might be on account of the Bpa photocrosslinker's preference for certain amino acids, propensity to form intramolecular rearrangement products, and that its crosslinking efficiency can be influenced by geometric constraints (Preston & Wilson, 2013).

### PhotoABPs specifically report RNF12 E3 ubiquitin ligase activity in vitro and in situ of cell extracts

We next set out to confirm mechanism-based labelling of RNF12 by photoABP. Under conditions where RNF12 was effectively labelled by photoABP, no labelling was observed with a photoABP carrying a F62A mutation which impairs E2 binding to the proximal RING domain (Zheng et al, 2000; Mathur et al, 2020) (Fig 2A). Furthermore, no labelling of RNF12 was observed with UbBpa31 where the E2 of the photoABP had been removed, illustrating the importance of this component of the photoABP (Mathur et al, 2020) (Fig 2B). Taken together, these data confirm that RNF12 probe labelling via the

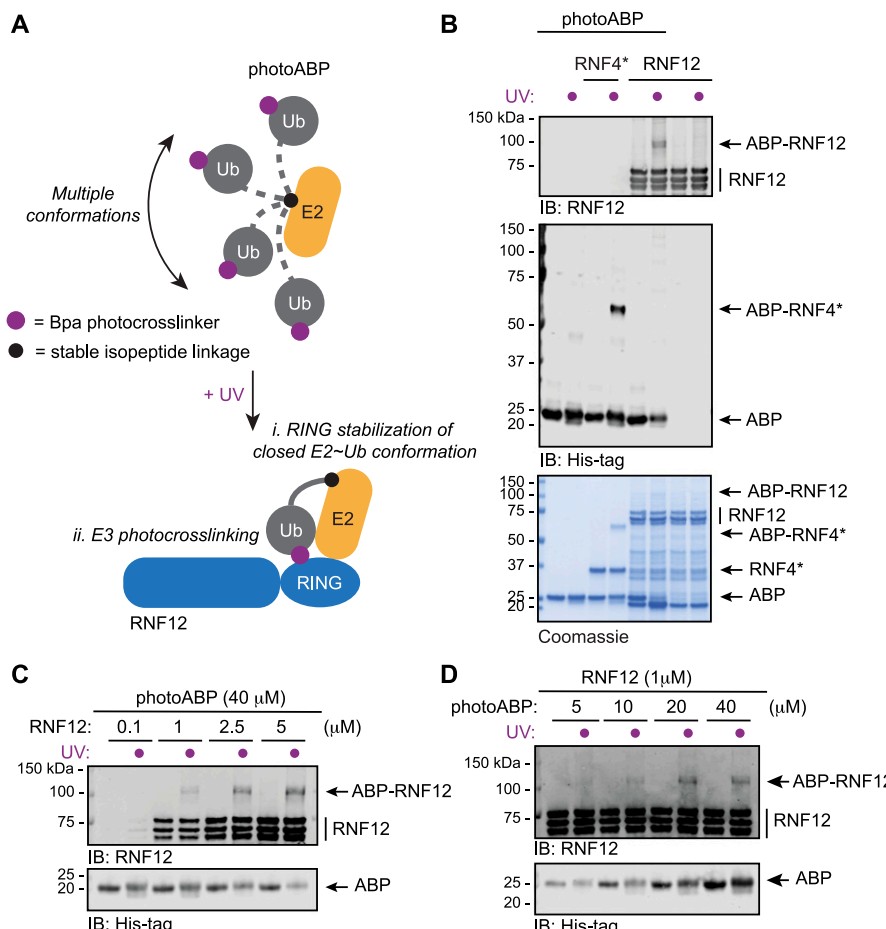

**Figure 1. Monitoring activity of the disease-associated RNF12 E3 ubiquitin ligase using a RING-specific photocrosslinking activity-based probe (photoABP).**

**(A)** Schematic of activity-dependent labelling of RNF12 by photoABP. The photoABP consists of a ubiquitin (Ub)-loaded E2 conjugating enzyme in which the E2~Ub bond is replaced by a stable lysine isopeptide (black dot) and a photoactivatable *p*-benzoyl-L-phenylalanine (Bpa) crosslinker (purple dot) is introduced. Like their native counterpart, photoABPs exist in a dynamic "open" E2-Ub conformational state in the absence of an active E3 ubiquitin ligase (top). However, engagement of an active RING E3 stabilises the photoABP "closed" E2-Ub conformation that is primed for Ub transfer, whereupon UV leads to photocrosslinking to the RING domain of RNF12 (bottom). **(B)** Recombinant RNF12 or RNF4* (5 μM) were incubated with photoABP (10 μM) and UV irradiated before analysis via immunoblotting with the indicated antibodies (top) or Coomassie staining (bottom). RNF4* is an RNF4 variant engineered to be constitutively active (Plechanovová et al, 2011). The experiment was repeated three times with similar results. Increased electrophoretic mobility of the photoABP in lane 6 is presumed to be due to RNF12 and UV induced intramolecular crosslinking. **(C, D)** Fixed amounts of recombinant RNF12 or photoABP, respectively, were incubated with increasing RNF12 (C) or photoABP (D) concentrations and UV irradiated before analysis via immunoblotting. Both experiments performed once.
Source data are available for this figure.

photocrosslinker in the ubiquitin component is dependent on RING engagement of the E2, consistent with the ubiquitylation mechanism used by RING E3s. Indeed, we confirmed that the isolated recombinant RNF12 RING domain (RNF12$^{RING}$) is sufficient to mediate crosslinking with photoABP (Fig 2C), and is labelled in a similar manner to RNF4*, but to a lesser extent (Fig 2C). This is consistent with the relative autoubiquitylation activities of isolated recombinant RNF12 RING domain and RNF4 RING domains ([Bustos et al, 2018], Fig 2D).

To directly demonstrate that photoABP can report disruptions to RNF12 E3 ubiquitin ligase activity, we used mutations in the RNF12 RING domain that have been shown to disrupt catalysis (Gontan et al, 2012; Bustos et al, 2018, 2020). The RNF12 H569A C572A double mutant is predicted to structurally disrupt the RING domain (Gontan et al, 2012), whereas RNF12 W576Y is predicted to interfere with the E2-RING interaction (Joazeiro, 1999; Das et al, 2013; Hodson et al, 2014) (Fig 3A). As expected, we confirmed that these mutants are inactive in single turnover lysine E2 ubiquitin discharge assays, which allows RING E3 catalytic activity to be quantified independent of a protein substrate (Fig 3B). Thus, we used these RING mutants as a benchmark of RNF12 activity for the photoABP assay. Compared with WT control, photoABP labelling of recombinant RNF12 W576Y and RNF12 H569A C572A was undetectable (Fig 3C). These data thereby confirm that RNF12 activity monitoring via photoABP is

dependent on engagement of the E2 conjugating enzyme component by an intact RING domain.

We next sought to test whether photoABP can be used to monitor RNF12 E3 ubiquitin ligase activity in situ of cell extracts. As a model system we reconstituted RNF12 knockout (*Rlim⁻/ʸ*) mouse embryonic stem cells (mESCs) (Bustos et al, 2018) with WT or RNF12 RING mutants introduced above (Fig 3A). Immunoblotting analysis confirms that RNF12 RING domain mutations impaired substrate ubiquitylation in mESCs, as measured by accumulation of the REX1 substrate (Fig 3D). Furthermore, RNF12 RING domain mutations disrupted RNF12-dependent *Usp26* gene expression, as measured by USP26 induction (Fig 3D), which we have previously shown is a transcriptional target of the RNF12-REX1 axis (Segarra-Fas et al, 2020 *Preprint*). Consistent with our in vitro data, UV-induced RNF12 crosslinking with photoABP was observed in mESC lysates expressing WT RNF12 (Fig 3E). However, RNF12 photoABP labelling was below the detection limit in mESC lysates expressing RNF12 H569A C572A or W576Y mutants (Fig 3E). Again, the minor fraction of RNF12 that is probe-labelled is likely due to factors that compromise the efficiency of the crosslinking reaction, rather than being reflective of the absolute stoichiometry of active RNF12. Taken together, these data indicate that the photoABP approach sensitively correlates with RNF12 activity in vitro and in situ of mammalian cell extracts.

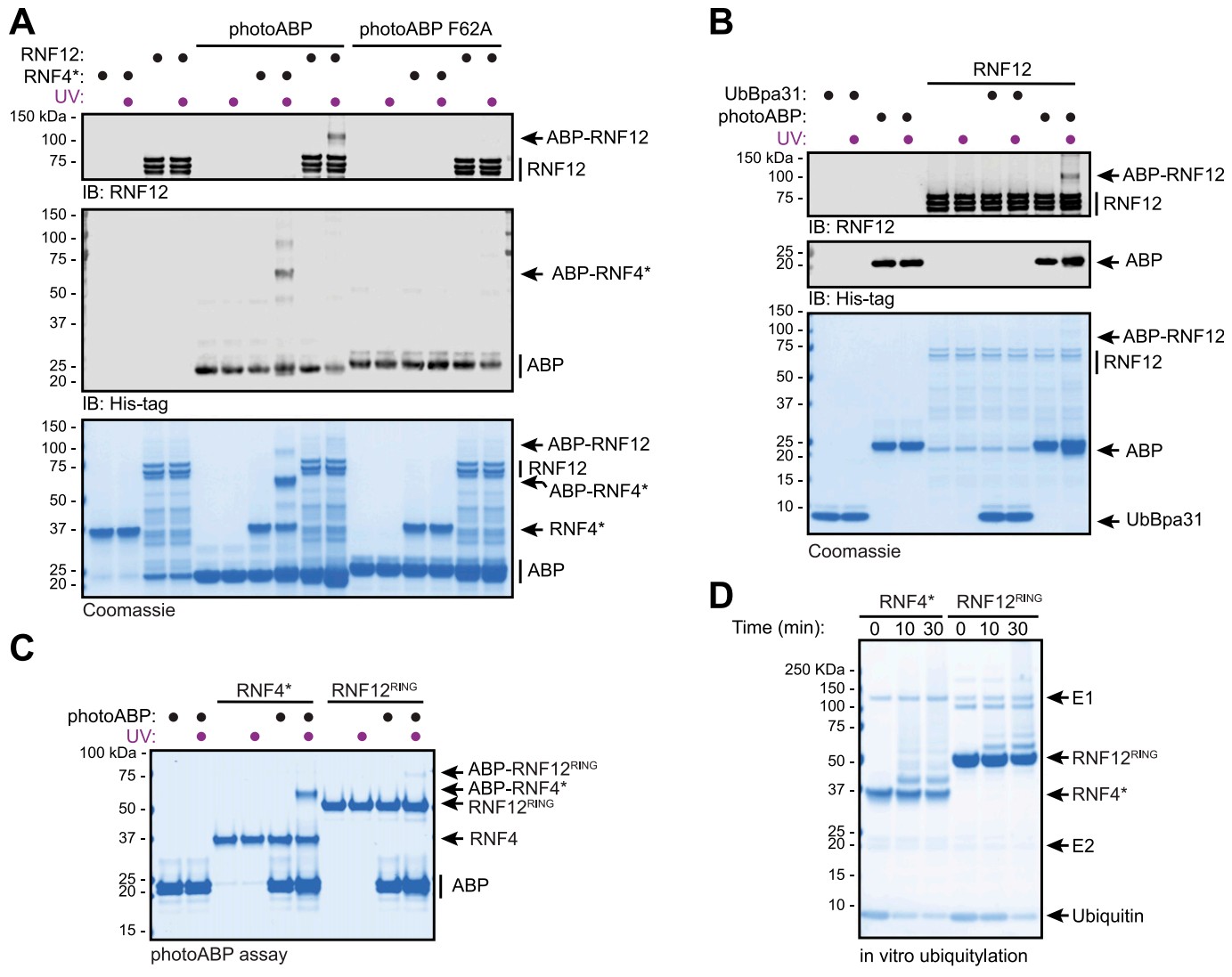

**Figure 2. RNF12 labelling with photoABP requires interaction with the integrated photoABP E2 conjugating enzyme.**
**(A)** Recombinant RNF12 or RNF4* were incubated with WT or F62A mutant photoABP and UV irradiated before analysis via immunoblotting with the indicated antibodies (top) or Coomassie staining (bottom). The experiment was repeated three times with similar results. **(B)** Recombinant RNF12 was incubated with photoABP or UbBpa31 and UV irradiated before analysis via immunoblotting with the indicated antibodies (top) or Coomassie staining (bottom). Experiment repeated twice with similar results. **(C, D)** Recombinant RNF12$^{RING}$ or RNF4* were subjected to a photoABP UV crosslinking assay (C) or an in vitro ubiquitylation assay for the indicated times (D) and samples analysed via Coomassie staining. Both experiments repeated twice with similar results.
Source data are available for this figure.

## PhotoABP monitors impact of pathogenic RNF12 variants on catalytic activity

Most reported RNF12 TOKAS variants are located in the RING domain and a distal basic region of the protein (Fig 4A) (Tønne et al, 2015; Hu et al, 2016; Frints et al, 2019; Bustos et al, 2021, 2018). Variants in both regions have been shown to disrupt catalytic activity, although basic region variants show milder defects compared with RING domain variants (Bustos et al, 2018; Frints et al, 2019). However, it is not known whether RNF12 RING or basic region variants correlate with the severity of TOKAS symptoms (Hu et al, 2016; Frints et al, 2019). Furthermore, the role of the RNF12 basic region in catalysis and the mechanism by which basic region variants disrupt RNF12 catalytic activity are not fully understood. Thus, we sought to determine whether photoABP can be used to quantify the disruption to RNF12 E3 activity caused by TOKAS variants and if this correlates with native RNF12 E3 activity. To this end, we assessed RNF12 activity of representative RNF12 R387C and R599C TOKAS variants in the basic region and RING domain, respectively, by single turnover lysine E2 ubiquitin discharge assay. As shown previously, RNF12 promotes lysine E2 ubiquitin discharge over time, which is similarly disrupted by RNF12 R387C and R599C TOKAS variants (Figs 4B and S1A and B, [Bustos et al, 2018]). These data suggest that both the RING domain and basic region serve important roles in stabilizing the active "closed" RING-E2-ubiquitin conformation, a function that is disrupted by TOKAS

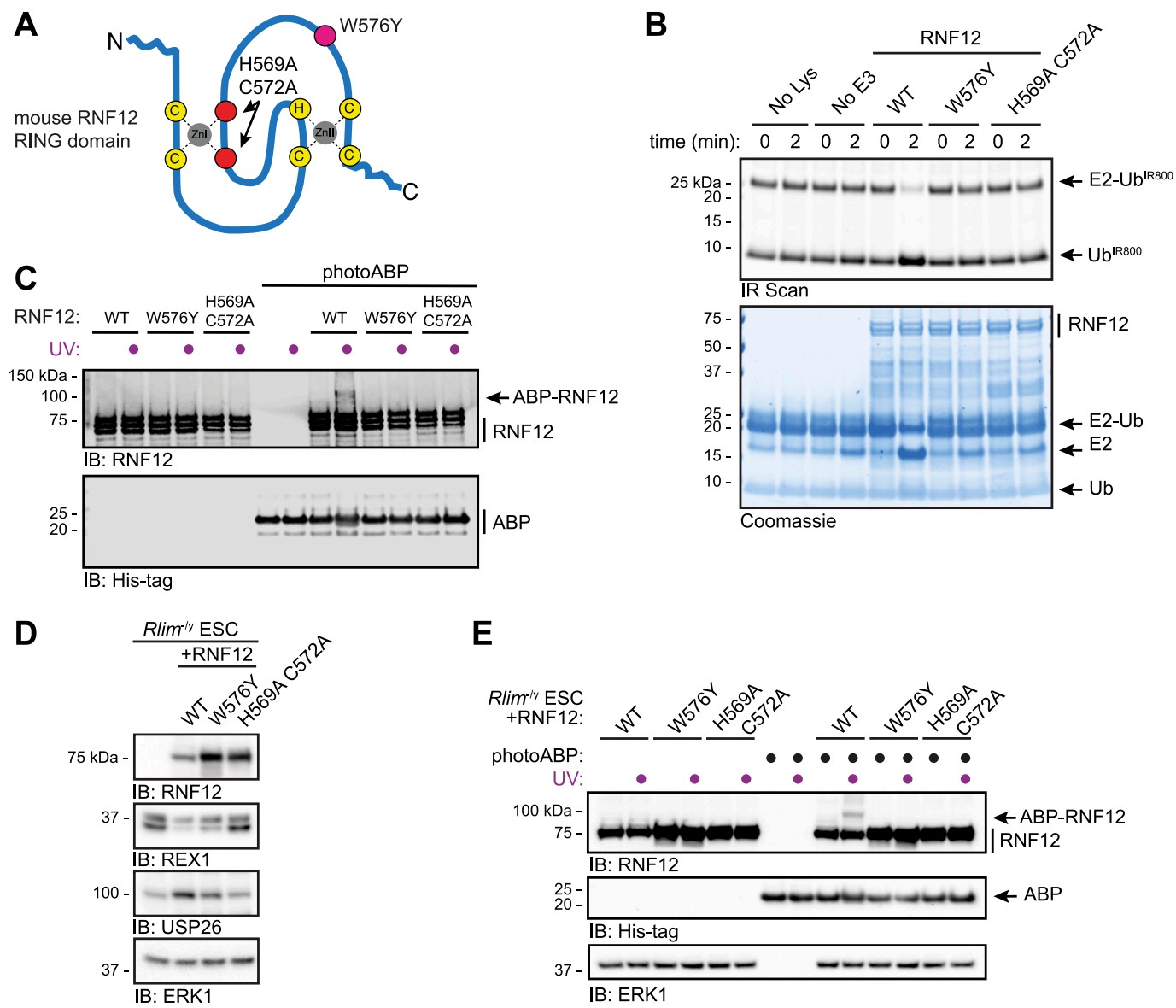

**Figure 3. PhotoABP activity reporting requires a functional RNF12 RING domain.**
**(A)** Diagram of the RNF12 RING domain indicating location of the inactivating mutations H569A C572A (red) and W576Y (fuchsia). **(B)** Recombinant RNF12 WT, W576Y, and H569A C572A were subjected to a fluorescent ubiquitin (Ub$^{IR800}$) E2 lysine discharge assay and RNF12-catalysed Ub$^{IR800}$ discharge from loaded UBE2D1 analysed via SDS–PAGE and infra-red (IR) scanning to visualise fluorescent Ub$^{IR800}$. Coomassie staining is shown as loading control. Experiment repeated twice with similar results. **(C)** Recombinant WT, W576Y, and H569A C572A mutant RNF12 were incubated with photoABP and UV irradiated before analysis via immunoblotting. Note that one reaction mix per condition (±UV) is divided in two to ensure that identical amounts of probe are added. The broadening of the photoABP band in lane 10 is presumed to be due to RNF12 and UV induced intramolecular crosslinking. **(D)** RNF12-deficient (*Rlim$^{-/y}$*) mouse embryonic stem cells (mESCs) were transfected with plasmid expressing the indicated RNF12 mutants before analysis via immunoblotting. ERK1 levels are shown as a loading control. **(E)** Cell lysates generated as in (D) were incubated with photoABP and UV irradiated before immunoblotting analysis with the indicated antibodies. ERK1 levels are shown as loading control. The experiments (C, D, E) were repeated four times with similar results.
Source data are available for this figure.

variants. Furthermore, we identify the basic region as a potential non-RING element (NRE; [Dou et al, 2013]) that plays a critical role in RNF12 catalysis.

Next, we quantified photoABP labelling of RNF12 and the TOKAS variants. To improve detection of the low activity demonstrated by these TOKAS variants with photoABP we increased labelling efficiency by using a high-power 365 nm LED lamp (UHP-T-365-MP,

Prizmatix). Under these conditions we find that labelling efficiency broadly correlated with the rate of lysine discharge, suggesting that photoABP analysis can serve as a reliable measure of native RNF12 E3 activity (Figs 4C and S1C). However, despite these readouts being mechanistically informative, a caveat is that they might not strictly correlate with ubiquitylation of a protein substrate (Bustos et al, 2018).

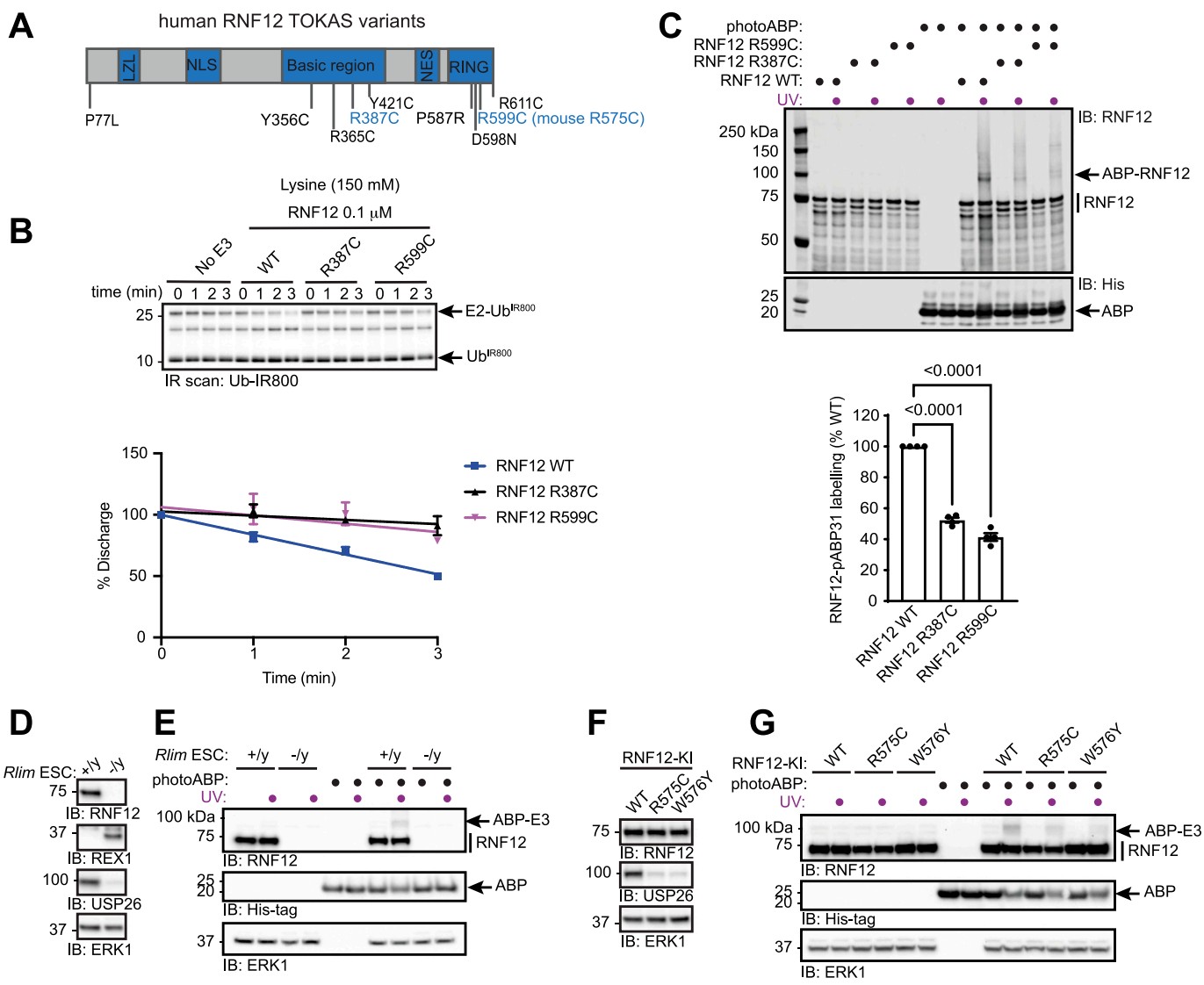

**Figure 4. PhotoABP measures disruptions in RNF12 E3 ubiquitin ligase activity caused by TOKAS variants.**
**(A)** Diagram of human RNF12 protein structure showing functional regions and positions of identified TOKAS variants. R387C and R599C are highlighted in blue. LZL, predicted leucine zipper-like; NLS, nuclear localisation signal; NES, nuclear export signal; RING, catalytic RING domain. **(B)** Recombinant human WT, R387C, and R599C RNF12 were subjected to a lysine E2 fluorescent ubiquitin ($Ub^{IR800}$) discharge assay for the indicated times and analysed via SDS–PAGE and in-gel infra-red (IR) scanning to visualise fluorescent $Ub^{IR800}$. Gel represents one of three replicates and is Coomassie stained in Fig S1A. Discharge was quantified and plotted as percentage of time = 0 min. Data are represented as biological replicates (n = 3). **(C)** Recombinant human RNF12 WT, R387C, and R599C were incubated with photoABP and UV irradiated before analysis via immunoblotting. For Coomassie stain see Fig S1C. Note that one reaction mix per condition (±UV) is divided in two to ensure that identical amounts of probe are added. RNF12 probe labelling was quantified and plotted as percentage of RNF12 WT. Data are represented as biological replicates (n = 4). **(D)** RNF12-expressing ($Rlim^{+/y}$) and RNF12-deficient ($Rlim^{-/y}$) mESC extracts were analysed via immunoblotting. ERK1 levels are shown as a loading control. **(E)** mESC extracts were incubated with photoABP and UV irradiated before analysis via immunoblotting. ERK1 levels are shown as loading control. **(D, E)** repeated twice with similar results. **(F)** RNF12 WT, R575C, or W576Y CRISPR/Cas9 knock-in (KI) mESCs were lysed and protein expression analysed via immunoblotting. ERK1 levels are shown as a loading control. **(G)** Samples generated as in (F) were incubated with photoABP, UV irradiated, and analysed via immunoblotting with the indicated antibodies. ERK1 levels are shown as a loading control. **(F, G)** repeated twice with similar results.
Source data are available for this figure.

As photoABPs report disruptions to RNF12 activity caused by the two major classes of TOKAS variants, we next set out to determine whether this approach might be used to monitor activity of RNF12 variants in cell lines harbouring patient-derived mutations. First, we tested the ability of photoABP to label endogenous RNF12 in RNF12-expressing ($Rlim^{+/y}$) and RNF12-deficient ($Rlim^{-/y}$) mESC extracts, which displayed REX1 substrate accumulation and impaired expression of RNF12-dependent genes such as *Usp26*, measured by USP26 protein levels (Fig 4D). Specific photoABP crosslinking of RNF12 was observed (Fig 4E), confirming that the photoABP can, in principle, report endogenous RNF12 catalytic activity.

We next sought to use photoABP to monitor the impact on catalytic activity of an endogenous RNF12 TOKAS variant. For this, we used CRISPR/Cas9 knock-in mESC lines harbouring RNF12 R575C

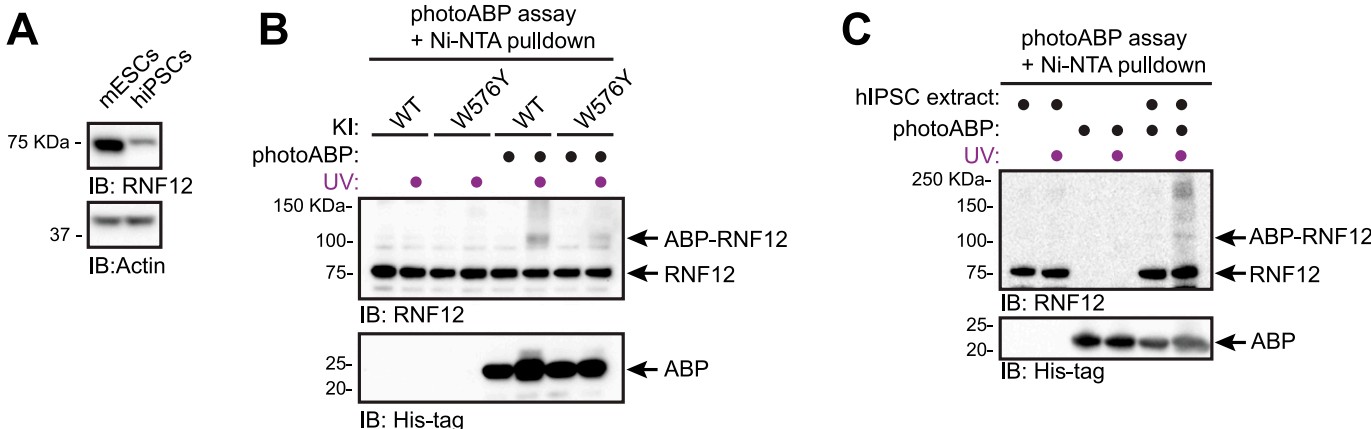

**Figure 5. PhotoABP reports RNF12 catalytic activity in human induced pluripotent stem cells (hiPSCs).**
**(A)** mESC or hiPSC extracts were analysed via immunoblotting. Actin levels are shown as loading control. **(B)** mESC extracts were incubated with photoABP and UV irradiated before Ni-NTA pulldown for photoABP-labelled proteins and immunoblotting analysis for RNF12 and photoABP (His-tag). Note the non-specific interaction between RNF12 and the resin, resulting in presence of unlabelled RNF12 in the Ni-NTA pull-down. **(C)** hiPSC extracts generated as in (A) were incubated with photoABP and UV irradiated before a Ni-NTA pulldown for photoABP-labelled proteins and immunoblotting analysis for RNF12 and photoABP (His-tag). The photoABP-RNF12 band in the absence of UV labelling is either low-level background probe labelling in the absence of UV or a non-specific band that is present at a lower level than the specific signal obtained for photoABP-RNF12 upon UV labelling. Note the non-specific interaction between RNF12 and the resin, resulting in presence of unlabelled RNF12 in the Ni-NTA pull-down. **(A, B, C)** repeated three times with similar results.
Source data are available for this figure.

(corresponding to human RNF12 R599C identified in TOKAS, see Fig 4A) alongside RNF12 WT control or W576Y knock-in mESC lines (Bustos et al, 2018, 2020). RNF12 R575C and W576Y mutations result in REX1 accumulation (Bustos et al, 2018; Segarra-Fas et al, 2020 *Preprint*) and reduced expression of the RNF12-responsive gene *Usp26*, again measured by USP26 protein levels (Fig 4F), confirming that these mutations impact on RNF12 activity. As expected, endogenous WT RNF12 was labelled by photoABP in these cell extracts. However, reduced RNF12 photoABP crosslinking was observed with R575C and W576Y knock-in mESC lysates (Fig 4G). Thus, photoABP reports severe loss of activity by an endogenously expressed RNF12 R575C TOKAS variant, consistent with previous results obtained through conventional ubiquitylation assays (Bustos et al, 2018). Our data therefore demonstrate that the photoABP approach can robustly monitor differences in RNF12 pathogenic variants in an endogenous cellular context.

### Using photoABP to monitor RNF12 activity in hiPSC samples

hiPSCs reprogrammed from somatic tissues have emerged as powerful tools for disease modelling and personalised medicine (Freel et al, 2020). Therefore, we used hiPSCs reprogrammed from dermal fibroblasts as a prototype human tissue-derived primary cell line for RNF12 photoABP activity profiling. As hiPSCs display lower levels of RNF12 expression compared with mESCs (Fig 5A, [Bustos et al, 2020]), we developed a sensitised version of the photoABP assay in which a Ni-NTA pulldown of the 6xHis-tag present in the photoABP probe is performed after the photo-crosslinking reaction. We confirmed that this assay specifically reports RNF12 activity when comparing WT with W576Y knock-in mESC extracts (Fig 5B). Importantly, RNF12 crosslinking was readily detected when performing the same photoABP assay in hiPSC extracts (Fig 5C). Therefore, these data indicate that the photoABP approach can be used to report RNF12

activity in human tissue-derived samples, and has potential for screening patient samples for defects in RNF12 activity caused by TOKAS and other as yet uncharacterised variants.

## Discussion

ABPs are rapidly gaining traction as valuable tools to interrogate the regulation of the ubiquitin system and its dysregulation in disease. In principle, ABPs have a major advantage over traditional biochemical assays as they can report changes in enzymatic activity of endogenous proteins in cells using limiting amounts of material from patient-derived samples. However, ABPs have not yet been deployed to monitor dysregulation in human disease of the largest class of enzymes in the ubiquitylation system, the RING E3 ubiquitin ligases. Here, we use a previously reported RING E3 ubiquitin ligase-specific photocrosslinking ABP (photoABP) to monitor catalytic activity of RNF12, an E3 ligase mutated in the human developmental disorder Tønne-Kalscheuer syndrome (TOKAS). We show that the photoABP specifically labels active RNF12 and sensitively detects disruptions in catalytic activity caused by RNF12 TOKAS variants, in vitro and in situ of cell extracts, including human tissue-derived induced pluripotent stem cells. This provides proof of principle for using photoABP technology to monitor dysregulated RING E3 activity in human diseases.

We envisage that RING photoABP technology can be used to rapidly determine the impact of RNF12 variants on catalytic activity. Importantly, this application is suitable for identification of distal variants located outside the core catalytic RING domain that nevertheless impact on catalysis. In that regard, we show that a TOKAS variant in the RNF12 basic region, which has poorly understood functions in catalysis, disrupts photoABP labelling. This indicates that

the RNF12 basic region is likely to play a key role in modulating the attainment of the "closed" E2-Ub-RING conformation, thereby providing further insight into the RNF12 catalytic mechanism.

We also propose that the photoABP will have utility to identify upstream signalling pathways that regulate RNF12 E3 activity and their disruption in disease. Indeed, we have recently shown that RNF12 is phosphorylated by Ser–Arg rich splicing factor protein kinase (SRPK), which drives catalytic activation and co-localisation with nuclear transcription factor substrates (Jiao et al, 2013; Bustos et al, 2020). Therefore, photoABP could be used to probe the function of SRPK and other kinase pathways in upstream regulation of RNF12 catalytic activity. Interestingly, SRPK family genes are also mutationally disrupted in developmental disorders characterised by intellectual disability (Bustos et al, 2020), suggesting that photoABP technology may be able to report in patients perturbations to upstream signalling pathways that regulate RNF12. Furthermore, in light of the current efforts in ubiquitin system drug discovery (Vassilev, 2007; Landré et al, 2014; Maculins et al, 2016; Quirit et al, 2017), it might be possible to exploit photoABP technology to streamline the screening for specific small molecule modulators of RNF12 E3 activity.

Finally, biomarker development is one of the biggest challenges for identifying underpinning causes of developmental disorders, which is crucial for effective therapeutic intervention (Symons & Roberts, 2013; Anderson, 2015; Zafarullah & Tassone, 2019). For this, a variety of potential biomarkers for intellectual disability disorders have been proposed including monitoring of heart activity (Tonnsen et al, 2013), blood cell signatures (Sengupta et al, 2019), metabolic (Berry-Kravis et al, 2015), inflammatory (Manti et al, 2018), and signalling molecules (Weng et al, 2008). Here, we show that the photoABP approach is a robust assay to monitor dysregulated RNF12 E3 activity in patients caused by TOKAS gene variants. We propose that the photoABP can constitute a robust biomarker principle for clinical screening purposes. This could also be applied to survey patients with overlapping clinical features caused by variants in other components of the pathway that affect RNF12 activity. In principle, photoABP assay is feasible in any cell type where RNF12 is expressed and can be easily established in clinical laboratories. Furthermore, our work provides proof of principle for using photoABP as a biomarker tool for a range of diseases including other developmental disorders where RING E3s such as MID2 (Geetha et al, 2014) and CHIP (Shi et al, 2014) have been implicated.

# Materials and Methods

## Preparation of isopeptide-linked photoABPs

Cloning, expression and purification of UbBpa31 and UBE2D3 (S22R C85K) have been described previously (Mathur et al, 2020). To generate the photoABP-Bpa31 probe, UBE2D3-containing S22R and C85K mutations (Plechanovová et al, 2012) (200 $\mu$M) was incubated with UbBpa31 (200 $\mu$M) and His6–Uba1 (1 $\mu$M) at 35°C for 26 h in conjugation buffer (50 mM Tris, pH 10.0, 150 mM NaCl, 3 mM ATP, 5 mM MgCl$_2$, and 1 mM TCEP). The E2–UbBpa31 conjugate was applied onto a HiLoad 16/60 Superdex 75 gel filtration column (GE Healthcare) (20 mM Hepes, pH 7.5, 150 mM NaCl, and 1 mM TCEP). The purified photoABP-Bpa31 probe was concentrated to 2 mg/ml and stored at

−80°C. A control probe containing an F62A mutation in the E2 component (photoABP F62A) was prepared using the same procedure.

## Recombinant proteins

Plasmids generated in pGEX-6P-1 backbone for expression in BL21 *Escherichia coli* were: mouse RNF12 (DU49041), mouse RNF12 H569A C572A (DU53251), and mouse RNF12 W576Y (DU58577), human RNF12 (DU53876), human RNF12 R387C (DU53877), and human RNF12 R599C (DU53898) and were purified by standard procedures by MRC-PPU Reagents and services. 6xHis MBP TEV mouse RNF12 K544-V600 RNF12$^{RING}$ (DU50971) was generated via the same procedures. These plasmids and recombinant proteins are available at http://mrcppureagents.dundee.ac.uk/. Constitutively active RNF4 composed of full-length RNF4 fused to a duplicate copy of the RING domain (RNF4*) was prepared as previously described (Plechanovová et al, 2011).

## Activity-based profiling of isolated RING E3 ligases

Photocrosslinking reactions (45 $\mu$l) were performed in a 24-well plate (Cryshem HR3-158; Hampton Research) in reaction buffer (20 mM Hepes, pH 7.5, 150 mM NaCl, and 1 mM TCEP) containing photoABP-Bpa31 probe (20 $\mu$M). Samples were divided into two portions. One portion was irradiated at 365 nm on ice at a distance of 2 cm from a handheld UV lamp (BLE-8T365; Spectroline), for 10–30 min and the other portion was preserved in the dark. Alternatively, for improved sensitivity a 365 nm LED lamp was used (UHP-T-365-MP; Prizmatix). Photocrosslinking reactions were terminated by addition of SDS–PAGE loading buffer and incubation of samples at 95°C for 5 min. For purified wild type, RNF12 concentrations were 1–5 $\mu$M, whereas for RNF12 mutants concentrations were 1 $\mu$M.

## Cell culture

CRISPR/Cas9 RNF12 knockout (*Rlim$^{-/y}$*) and WT, R575C, and W576Y RNF12 knock-in male mESCs were described previously (Bustos et al, 2018, 2020). Wild-type or CRISPR/Cas9-edited mESCs were grown on 0.1% gelatin (wt/vol)–coated plates in ES-DMEM media (DMEM supplemented with 10% (vol/vol) FBS, 5% (vol/vol) Knock-Out serum replacement, 20 ng/ml GST-tagged leukemia inhibitory factor, penicillin/streptomycin, 2 mM glutamine, 0.1 mM minimum essential media, non-essential amino acids, 1 mM sodium pyruvate (all Thermo Fisher Scientific), and 0.1 mM $\beta$-mercaptoethanol (Sigma-Aldrich)) at 5% CO$_2$ and 37°C.

mESC transfection was performed using Lipofectamine LTX (Thermo Fisher Scientific) according to the manufacturer instructions. Plasmids in a pCAGGS puro backbone were generated by MRC-PPU Reagents and Services (University of Dundee) and were mouse RNF12 (DU50610), RNF12 H569A C572A (DU50631), and RNF12 W576Y (DU50800). Plasmids can be obtained from MRC-PPU Reagents and services website http://mrcppureagents.dundee.ac.uk/.

CHiPS4 male hiPSCs were grown in TeSR medium supplemented with Noggin (10 ng/ml; Peprotech) and bFGF (30 ng/ml; Peprotech) on 20 mg/cm$^2$ Geltrex matrix coated plates (Life Technologies) at 5% CO$_2$ and 37°C.

### Activity-based RING E3 profiling in cell extracts

mESCs were homogenised in lysis buffer (50 mM $Na_2HPO_4$, 10 mM glycerophosphate, 50 mM sodium fluoride, 5 mM sodium pyrophosphate, 1 mM sodium vanadate, 0.25 M sucrose, 50 mM NaCl, 0.2 mM PMSF, 1 mM benzamidine, 10 $\mu$M TCEP, and 1% NP-40). PhotoABP-Bpa31 probe (20 $\mu$M) was mixed with 40–50 $\mu$g cell lysate and UV irradiated (10 min) using the photocrosslinking procedure described above. Reactions were stopped with 1× LDS SDS–PAGE loading buffer and samples boiled at 95°C for 5 min. Alternatively, photoABP-Bpa31 – lysate mixtures were UV irradiated for 30 min and subjected to a Ni-NTA pulldown procedure. Photocrosslinking reactions were diluted 50-fold in lysis buffer and incubated with Ni-NTA agarose beads on a rotating shaker at 4°C for 20 h. Beads were then washed thrice with lysis buffer and twice with TI buffer (25 mM Tris, pH 6.8, 20 mM imidazole) and eluted in elution buffer (25 mM Tris, pH 6.8, 300 mM imidazole, and 50 mM DTT), mixed to 1× LDS SDS–PAGE loading buffer, and boiled at 95°C for 5 min before immunoblot analysis.

### SDS–PAGE electrophoresis and immunoblotting

Samples were separated in 4–12% SDS–PAGE gels using MES running buffer and analysed via Coomassie staining using InstantBlue reagent (Abcam) and imaged in a Gel-Doc XR+ System (Bio-Rad) or document scanner. Alternatively, proteins were transferred to nitrocellulose membranes and analysed by immunoblotting with primary antibodies for RNF12 amino acids 1–271 (S691D third bleed, 1:1,000), USP26 (SA085, third bleed, 1:1,000), both from MRC-PPU Reagents and Services, ERK1 (1:1,000; BD), REX1 (1:1,000; Abcam), Actin (Cell Signaling Technologies), and His-tag (1:10,000; Clontech). After secondary antibody incubation, protein signals were analysed by chemiluminescence detection with Immobilon Western Chemiluminescent HRP substrate (Millipore) using a Gel-Doc XR+ System (Bio-Rad) or by Infrared detection using an LI-COR Odyssey CLx system.

### Fluorescent E2~ubiquitin lysine discharge assays

UBE2D1-ubiquitin thioester was prepared by incubating 25 $\mu$M UBE2D1, 0.2 $\mu$M UBE1 (MRC-PPU Reagents and Services, University of Dundee), and 2 $\mu$M DyLight 800 Maleimide fluorescently labelled ubiquitin (Ub-IR[800]) (Kumar et al, 2015) in reaction buffer (50 mM Tris pH 7.5, 25 $\mu$M ubiquitin, 3 mM ATP, 0.5 mM TCEP [all from Sigma-Aldrich], 5 mM $MgCl_2$, and 150 mM NaCl) for 20 min at 37°C. Reaction was stopped by addition of MLN4924 derivative, compound 1 (25 $\mu$M), which inhibits E1 (Brownell et al, 2010; Pao et al, 2018) for 15 min at RT. Then, UBE2D1-ubiquitin conjugates were incubated with 100 nM RNF12 and 150 mM L-lysine in a buffer containing 50 mM Tris, pH 7.5, 150 mM NaCl, 0.5 mM TCEP, and 0.1% (vol/vol) NP40 at RT. Reaction was stopped by addition of non-reducing SDS loading buffer and samples were analysed via SDS–PAGE and in-gel fluorescence was measured with a Gel-Doc XR+ System (Bio-Rad). Gels were then stained with Coomassie (InstantBlue reagent; Abcam) and scanned in a Gel-Doc XR+ System (Bio-Rad).

### In vitro ubiquitylation assays

Isolated RING ubiquitylation assays were performed by incubating 5 $\mu$M E3 (RNF12[RING] or RNF4*) with a ubiquitylation reaction mix containing 0.1 $\mu$M UBE1, 0.5 $\mu$M UBE2D1 (both from MRC-PPU Reagents and Services), 5 $\mu$M Ubiquitin, 5 mM ATP, and 0.5 mM TCEP (all from Sigma-Aldrich), for 30 min at 30°C. Reactions were stopped with SDS–PAGE loading buffer and boiled at 95°C for 5 min. Samples were separated via SDS–PAGE electrophoresis and gels were stained with Coomassie (InstantBlue reagent; Abcam) and scanned in a Gel-Doc XR+ System (Bio-Rad).

## Data Availability

All data from this publication are available within the Figures, Supplementary Material and Source Data provided.

## Supplementary Information

## Acknowledgements

We thank MRC-PPU Reagents and Services, University of Dundee, for cDNA cloning, antibody and protein production, Dr. Viduth Chaugule and Dr. Helen Walden, University of Glasgow for Ub-IR[800], and Dr. Lindsay Davidson, Pluripotent Stem Cell Facility, School of Life Sciences, University of Dundee for hiPSCs. We thank Ron Hay, University of Dundee, for the constitutive RNF4 clone. We acknowledge funding from companies supporting the Division of Signal Transduction Therapy (Boehringer-Ingelheim, GlaxoSmithKline and Merck KGaA). GM Findlay and F Bustos are supported by a Wellcome Trust/ Royal Society Sir Henry Dale Fellowship (211209/Z/18/Z), a Tenovus Scotland Research Grant (T19/25) and a Medical Research Council New Investigator Award (MR/N000609/1). S Virdee and S Mathur are supported by United Kingdom MRC (MC_UU_12016/8) and the Biotechnology and Biological Sciences Research Council (BB/P003982/1). C Espejo-Serrano is supported by a Wellcome Trust PhD studentship.

### Author Contributions

F Bustos: conceptualization, data curation, formal analysis, validation, investigation, visualization, methodology, and writing—original draft, review, and editing.
S Mathur: conceptualization, validation, data curation, investigation, methodology, formal analysis, and writing—review and editing.
C Espejo-Serrano: data curation, formal analysis, validation, and investigation.
R Toth: resources and supervision.
CJ Hastie: resources and supervision.
S Virdee: conceptualization, resources, supervision, funding acquisition, and writing—original draft, review, and editing.
GM Findlay: conceptualization, resources, data curation, formal analysis, supervision, funding acquisition, visualization, project administration, and writing—original draft, review, and editing.

## Conflict of Interest Statement

S Virdee and S Mathur are authors of a patent involving the ABP technology and S Virdee is founder and shareholder of Outrun Therapeutics, a biotech company focused on the ubiquitin system.

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
