## [Reviewer comments · Life Science Alliance]

Life Science Alliance

ACTIVITY-BASED PROBE PROFILING OF RNF12 E3 UBIQUITIN LIGASE FUNCTION IN TONNE-KALSCHEUER SYNDROME

Francisco Bustos, Sunil Mathur, Carmen Espejo-Serrano, Rachel Toth, C. James Hastie, Satpal Virdee, and Greg Findlay
DOI: <https://doi.org/10.26508/lsa.202101248>

Corresponding author(s): Greg Findlay, MRC Protein Phosphorylation and Ubiquitylation Unit and Satpal Virdee, MRC Protein Phosphorylation and Ubiquitylation Unit

Review Timeline:

Submission Date:	2021-09-28
Editorial Decision:	2021-10-25
Revision Received:	2022-04-14
Editorial Decision:	2022-05-03
Revision Received:	2022-05-30
Editorial Decision:	2022-06-01
Revision Received:	2022-06-08
Accepted:	2022-06-08

Transaction Report:

October 25, 2021

Re: Life Science Alliance manuscript #LSA-2021-01248

Dr. Greg Findlay
MRC Protein Phosphorylation and Ubiquitylation Unit
School of Life Sciences
Dundee DD1 5EH
United Kingdom

Dear Dr. Findlay,

Thank you for submitting your manuscript entitled "ACTIVITY-BASED PROBE PROFILING OF RING E3 UBIQUITIN LIGASE FUNCTION IN A DEVELOPMENTAL DISORDER" to Life Science Alliance. The manuscript was assessed by expert reviewers, whose comments are appended to this letter. We invite you to submit a revised manuscript addressing the Reviewer comments.

Thank you for this interesting contribution to Life Science Alliance. We are looking forward to receiving your revised manuscript.

Sincerely,

B. MANUSCRIPT ORGANIZATION AND FORMATTING:

Reviewer #1 (Comments to the Authors (Required)):

In this manuscript by Bustos and Mathur, et al., a recently developed activity-based probe for RING-type E3 ligases was used to profile ligase defects in RNF12 caused by mutations associated with Tonne-Kalscheuer syndrome. The authors present well-controlled experiments showing that wild-type RNF12 can react (though very modestly) with the photoABP, and that this reactivity can report on the defects attributed to both synthetic/structure-guided mutations (W576Y, H569A/C572A) as well as disease-associated (R387C and R599C) mutations, both in and outside of the RNF12 RING domain itself. This readout on RNF12 activity is demonstrated for recombinant protein, material expressed in transfected mESCs, as well as material endogenously expressed in hiPSCs. The technique is a major advance from previous approaches that may have relied upon enrichment or recombinant expression/purification of RNF12, both of which take the ligase out of its native context. A limitation of this work, however, is that application of the photoABP requires cell lysis as well as prolonged UV exposure, which in some instances may skew results. This work, however, represents an important advance for profiling the effects of mutations in RING E3 ligases and will be of great interest to those among the LSA readership that are interested in protein ubiquitination and/or genetic diseases.

Major concerns:

- 1) It is unclear how sensitive the photoABP readout is to small differences in RNF12 activity. For example, Fig. 3D suggests that the W576Y mutant has some residual activity in cells, but the photoABP reports no observable activity in Fig. 3E. Likewise for the patient mutations R387C and R599C, some residual activity of the recombinant protein is evident in Fig. 4B, but no photoABP reactivity is seen in Fig. 4C. Thus, the photoABP readout for activity appears to be much more "black and white", and does not provide the same insight as the conventional ubiquitination assays presented in Bustos et al., 2018, which report no activity for the R599C mutant and an intermediate effect for the R387C mutant. Can the authors explain this discrepancy and, if this is a limitation of the approach, discuss the matter further in the text?
- 2) In Fig. 5C, the authors present an approach to test native RNF12 from hiPSC cells, which required enrichment of photoABP-reacted material prior to detection of the reacted species by Western blot. Firstly, can the authors explain why there is so much residual unmodified RNF12 that survives the Ni-NTA pulldown? And secondly, the gel band that the authors point to as photoABP-RNF12 is clearly also present in the control sample that did not receive UV photoactivation; can the authors explain the presence of this species?

Minor concerns:

- 1) The authors refer to Fig. 2D as evidence of "full-length RNF12 and RING domain truncations demonstrating comparable activity in in vitro ubiquitylation assays", but this figure panel actually compares the activities of RNF4 and RNF12, not the two constructs of RNF12.
- 2) It was not apparent how many times the authors replicated their results.

Reviewer #2 (Comments to the Authors (Required)):

The paper by Bustos and colleagues describes the use of an activity-based probe (photoABP) to follow the activity of the RING ligase RNF12 and pathogenic variants of this E3. Originally reported in a previous study where the probe was used to characterise the RNF4 and Cbl RING ligases, the authors now show the extended use of this probe to label the active form of RNF12, both in vitro and in cell lysates. Furthermore, authors use photoABP to monitor the influence of pathogenic variants of RNF12, which disrupt its E3 ligase activity.

The paper is well written and describes an interesting extended use of photoABP to monitor the dysregulation of ubiquitination in a pathogenic context. However, some of the claims made about the general applicability of the photoABP to specifically monitor the activity of RNF12 and its mutants, especially in a cellular context, are not fully supported by the data shown and the authors should rephrase and weaken some of their conclusions.

Specific points:

1. Although the authors provide extended experimental characterisation of the activity of the photoABP towards RNF12, the labeling efficiency is rather low and it is not clear how reproducible it is. For example, in Figure 1B, the photoABP-RNF12 band is only visible by immunoblotting against RNF12, but not by probing for the photoABP (via His), which clearly shows labeling of RNF4-RING, which serves as a positive control. Similar behaviour is seen when authors compare the activity of the isolated RING domains of RNF12 and RNF4-RING (Figure 2C). The authors claim that both are "labeled to a similar extent", which is not supported by the data shown. This is especially obvious on the Coomassie gel shown in Figure 2A. This suggests that, although both proteins are similar in activity (Figure 2D), the photoABP is not as suitable to monitor RNF12 activity as it is for RNF4. The authors should comment on this.
2. The low labeling efficiency also raises issues with the general application of this approach to investigate the specific effect of mutations on activity, by monitoring a decrease/absence of labeling. Based on the low signal in western blots, this reviewer is not convinced that differences in activity levels can be reliably detected. To support this claim the authors should carry out a time course of lysine discharge assays with WT and mutants, quantify the data and compare them to quantified labelling efficiency.
3. While this probe is a very interesting and valuable addition to the arsenal of ubiquitin probes available, the authors should not overstate the claims regarding its use. For example, on page 5 the authors state: "providing new mechanistic insights into how TOKAS variants disrupt catalytic activity." In this paper the photoABP is only used to monitor the presence or absence of activity of RNF12, without providing insight into a molecular mechanism. Similarly, the authors suggest that the photoABP can be used to "robustly monitor differences in RNF12 activity" (page 15), but this is not supported by data, as the low labeling efficiency is unlikely to monitor differences in activity robustly.
4. The authors claim throughout the manuscript that the ABP has been tested in patient-derived cell lines. This is misleading because hiPSCs are not cells from TOKAS patients, although they are of human origin. By making this claim, the authors are exaggerating the extent to which their ABP has been used in disease models. This is alluded to by calling these cells a 'prototype patient-derived cell line'. In addition, it should be made clear in the manuscript that ABP incubations were performed in cell extracts; the repeated use of 'in ... cell-lines' suggests that incubations were performed in intact cells.
5. In Figure 4G there is a band visible in the WT non-UV lane without the probe at the same position as the probe labelling. Could the authors comment on the presence of non-specific labeling?
6. In Figures 5B and 5C, why is RNF12 detected in the NiNTA pull-down samples? The pull-down does not seem to enrich photoABP-RNF12, compared to assays without this step (Figure 4G). The authors should describe in the figure legend how this experiment was carried out. There also seems to be a band indicating labeling of the W576Y mutant.
7. In some experiments, the signal for the photoABP in cell lysate assays appears stronger in the RNF12 wild-type sample compared to mutants (Figure 3C; Figure 4C). Did authors use more photoABP in these assays?

Reviewer #3 (Comments to the Authors (Required)):

In their manuscript, Bustos et al. tested the hypothesis if RNF12 RING E3 ubiquitin ligase activity can be assessed by photocrosslinking Activity-Based Probes (photoABPs). They used this assay to investigate if photoABP binding to RNF12 differs in RNF12 mutants that are associated with the developmental disorder Tonne-Kalscheuer syndrome (TOKAS) and found this to be reduced. As a proof of principle this method was used in patient derived cell-lines, however, not from TOKAS patients. The manuscript is well prepared and the experiments are overall well controlled. However, I do have some points that should be addressed before the manuscript is ready for publication.

Major comments:

The E2's UBCH5A [1, 2] and UBC8 [3] are known to cooperate with RNF12. Please explain why UBE2D3 (S22R C85K) and not UBCH5A or UBC8 has been used in the assay. The paper where UBE2D3 was shown to cooperate with RNF12 for substrate degradation needs to be cited. Otherwise it needs to be stated that the assay relies on an unspecific E3 RING - E2 interaction. Several targets of RNF12, such as REX1[1], HDAC2 [3], TRF1 [2], Stathmin [4], ERalpha [5], BRF1 [6] and SMAD7 [7] were described in the literature. At least some of the known RNF12 substrates should be mentioned in the manuscript and a reason why REX1 was chosen as RNF12 substrate needs to be provided.

For a broader readership background information for RNF12 is required. Please describe whether or not RNF12 is expressed in a tissue specific manner, if its expression or content is changed during development and in which cellular compartment it is contained.

On page 13 the following is stated "Furthermore, RNF12 RING domain mutations disrupt RNF12-dependent gene expression, as measured by USP26 induction." However, USP26 protein is determined in the assays. In order to state that RNF12-dependent gene expression is changed USP26 expression needs to be measured and reported.

The title should be focused on the content of the manuscript; namely the usage of photoABPs to analyze RNF12 TOKAS variants.

Minor comments:

For the general audience the purpose of the "single turnover lysine E2 ubiquitin discharge assay" needs to be explained.

Page 4: Figure 1A does not show what the authors refer to.

Figures 3 and 4: Please explain why ERK1 immunoblots were performed.

Figure 3 and 4: please indicate what "IR scan" stands for.

Figure 4F: please show immunoblot for REX1

Figure 4G: please explain the signal in the first lane in IB: RNF12, which migrates at the heights of photoABP-RNF12, and leave questions about the specificity of the assay.

References

1. Gontan C, Achame EM, Demmers J, Barakat TS, Rentmeester E, van IW, et al. RNF12 initiates X-chromosome inactivation by targeting REX1 for degradation. *Nature*. 2012;485:386-90. doi:10.1038/nature11070
2. Her YR, Chung IK. Ubiquitin Ligase RLIM Modulates Telomere Length Homeostasis through a Proteolysis of TRF1. *J Biol Chem*. 2009;284:8557-66. doi:10.1074/jbc.M806702200
3. Kramer OH, Zhu P, Ostendorff HP, Golebiewski M, Tiefenbach J, Peters MA, et al. The histone deacetylase inhibitor valproic acid selectively induces proteasomal degradation of HDAC2. *EMBO J*. 2003;22:3411-20. doi:10.1093/emboj/cdg315
4. Chen X, Shen J, Li X, Wang X, Long M, Lin F, et al. Rlim, an E3 ubiquitin ligase, influences the stability of Stathmin protein in human osteosarcoma cells. *Cell Signal*. 2014;26:1532-8. doi:10.1016/j.cellsig.2014.03.018
5. Johnsen SA, Gungor C, Prenzel T, Riethdorf S, Riethdorf L, Taniguchi-Ishigaki N, et al. Regulation of estrogen-dependent transcription by the LIM cofactors CLIM and RLIM in breast cancer. *Cancer Res*. 2009;69:128-36. doi:10.1158/0008-5472.CAN-08-1630
6. Wang F, Zhao K, Yu S, Xu A, Han W, Mei Y. RNF12 catalyzes BRF1 ubiquitination and regulates RNA polymerase III-dependent transcription. *J Biol Chem*. 2019;294:130-41. doi:10.1074/jbc.RA118.004524
7. Zhang L, Huang H, Zhou F, Schimmel J, Pardo CG, Zhang T, et al. RNF12 controls embryonic stem cell fate and morphogenesis in zebrafish embryos by targeting Smad7 for degradation. *Mol Cell*. 2012;46:650-61. doi:10.1016/j.molcel.2012.04.003

Reviewer #1 (Comments to the Authors (Required)):

In this manuscript by Bustos and Mathur, et al., a recently developed activity-based probe for RING-type E3 ligases was used to profile ligase defects in RNF12 caused by mutations associated with Tonne-Kalscheuer syndrome. The authors present well-controlled experiments showing that wild-type RNF12 can react (though very modestly) with the photoABP, and that this reactivity can report on the defects attributed to both synthetic/structure-guided mutations (W576Y, H569A/C572A) as well as disease-associated (R387C and R599C) mutations, both in and outside of the RNF12 RING domain itself. This readout on RNF12 activity is demonstrated for recombinant protein, material expressed in transfected mESCs, as well as material endogenously expressed in iPSCs. The technique is a major advance from previous approaches that may have relied upon enrichment or recombinant expression/purification of RNF12, both of which take the ligase out of its native context. A limitation of this work, however, is that application of the photoABP requires cell lysis as well as prolonged UV exposure, which in some instances may skew results. This work, however, represents an important advance for profiling the effects of mutations in RING E3 ligases and will be of great interest to those among the LSA readership that are interested in protein ubiquitination and/or genetic diseases.

Major concerns:

1) It is unclear how sensitive the photoABP readout is to small differences in RNF12 activity. For example, Fig. 3D suggests that the W576Y mutant has some residual activity in cells, but the photoABP reports no observable activity in Fig. 3E. Likewise for the patient mutations R387C and R599C, some residual activity of the recombinant protein is evident in Fig. 4B, but no photoABP reactivity is seen in Fig. 4C. Thus, the photoABP readout for activity appears to be much more "black and white", and does not provide the same insight as the conventional ubiquitination assays presented in Bustos et al., 2018, which report no activity for the R599C mutant and an intermediate effect for the R387C mutant. Can the authors explain this discrepancy and, if this is a limitation of the approach, discuss the matter further in the text?

We thank the reviewer for highlighting the issue of whether the probe is sufficiently sensitive to reveal changes in activity uncovered by conventional ubiquitylation assays and the discrepancy with substrate ubiquitylation data reported in Bustos et al. 2018. We have now optimised probe labelling of RNF12 mutants using a more efficient 365 nm LED lamp and provide new data demonstrating detectable probe labelling of RNF12 R599C and R387C (Figure 4C). We now provide a quantitative comparison of RNF12 lysine discharge activity (Figure 4B) with photoABP labelling efficiency (Figure 4B vs. 4C). These data confirm that results obtained from probe labelling and lysine discharge approaches are quantitatively similar, and consistent with previous data from Bustos et al. 2018. Although qualitative substrate ubiquitylation data in Bustos et al. 2018 indicates R387C is more active than R599C, the quantitative lysine discharge data presented in that paper, and in our revised manuscript, indicates the two mutants have similarly impaired activity. This is consistent with our new quantitative probe data.

We agree the discrepancy between lysine discharge/probe and substrate ubiquitylation should be highlighted and we have added text describing this caveat. Lysine discharge assays remain a valuable way of assaying RING E3 activity, albeit in the context of a model substrate. As the photoABP assay is highly versatile allowing direct measurement of cellular RNF12 activity, we believe it remains of great value.

“However, despite these readouts being mechanistically informative, a caveat is they might not strictly correlate with ubiquitylation of a protein substrate (Bustos et al, Cell Reports 2018).”

2) In Fig. 5C, the authors present an approach to test native RNF12 from hiPSC cells, which required enrichment of photoABP-reacted material prior to detection of the reacted species by Western blot. Firstly, can the authors explain why there is so much residual unmodified RNF12 that survives the Ni-NTA pulldown?

Unfortunately, unlabelled RNF12 binds non-specifically to the Ni-NTA resin. However, this does not impact on the specificity of the assay as labelled RNF12 is only detected in the presence of photoABP and UV-induced cross-linking (compare lanes, 2, 5 & 6). We explain the presence of this band in the legend for Figure 5C.

And secondly, the gel band that the authors point to as photoABP-RNF12 is clearly also present in the control sample that did not receive UV photoactivation; can the authors explain the presence of this species?

The photoABP-RNF12 band in the absence of UV labelling is either very low-level background probe labelling in the absence of UV or a non-specific band that is present at a much lower level than the specific signal obtained for photoABP-RNF12 signal upon UV labelling. We highlight this in the legend for Figure 5C.

Minor concerns:

1) The authors refer to Fig. 2D as evidence of "full-length RNF12 and RING domain truncations demonstrating comparable activity in in vitro ubiquitylation assays", but this figure panel actually compares the activities of RNF4 and RNF12, not the two constructs of RNF12.

We apologise to the reviewer for this error. The data in 2D is correct and consistent with our intended narrative. We have now corrected:

“In accordance both RNF4 and RNF12^{RING} demonstrated autoubiquitylation activity ((Bustos et al, 2018), Figure 2D).”*

N.B. For clarity we have changed the name of the constitutively active RNF4-RING construct to RNF4* and defined this appropriately.

2) It was not apparent how many times the authors replicated their results.

Many thanks for the reviewer for pointing this out, we have now added “n=” to the figure legends for quantitative data. For qualitative data we have specified the number of times experiments were repeated with similar results.

Reviewer #2 (Comments to the Authors (Required)):

The paper by Bustos and colleagues describes the use of an activity-based probe (photoABP) to follow the activity of the RING ligase RNF12 and pathogenic variants of this E3. Originally reported in a previous study where the probe was used to characterise the RNF4 and Cbl RING ligases, the authors now show the extended use of this probe to label the active form of RNF12, both in vitro and in cell lysates. Furthermore, authors use photoABP to monitor the influence of pathogenic variants of RNF12, which disrupt its E3 ligase activity.

The paper is well written and describes an interesting extended use of photoABP to monitor the dysregulation of ubiquitination in a pathogenic context. However, some of the claims made about the general applicability of the photoABP to specifically monitor the activity of RNF12 and its mutants, especially in a cellular context, are not fully supported by the data shown and the authors should rephrase and weaken some of their conclusions.

Specific points:

1. Although the authors provide extended experimental characterisation of the activity of the photoABP towards RNF12, the labeling efficiency is rather low and it is not clear how reproducible it is. For example, in Figure 1B, the photoABP-RNF12 band is only visible by immunoblotting against RNF12, but not by probing for the photoABP (via His), which clearly shows labeling of RNF4-RING, which serves as a positive control. Similar behaviour is seen when authors compare the activity of the isolated RING domains of RNF12 and RNF4-RING (Figure 2C). The authors claim that both are "labeled to a similar extent", which is not supported by the data shown. This is especially obvious on the Coomassie gel shown in Figure 2A. This suggests that, although both proteins are similar in activity (Figure 2D), the photoABP is not as suitable to monitor RNF12 activity as it is for RNF4. The authors should comment on this.

We thank the reviewer for this comment and agree that RNF4 and RNF12 are not "labeled to a similar extent". It was not our intention to mislead, and we have now corrected in the text. One likely reason the probe appears more suitable to measure RNF4 than RNF12 might be because the activities are not directly comparable. RNF4-RING (now referred to as RNF4* for clarity) is engineered to be constitutively active and consists of full length RNF4 fused to another RNF4 RING domain. In contrast, we do not know what fraction of RNF12 in our recombinant preps and in cells is active and the reduced probe labelling efficiency might be reflective of the majority being inactive. Another possibility is the residue(s) that are in proximity of the Bpa photocrosslinker in RNF12 undergo photocrosslinking with reduced efficiency compared to RNF4.

2. The low labeling efficiency also raises issues with the general application of this approach to investigate the specific effect of mutations on activity, by monitoring a decrease/absence of labeling. Based on the low signal in western blots, this reviewer is not convinced that differences in activity levels can be reliably detected. To support this claim the authors should carry out a time course of lysine discharge assays with WT and mutants, quantify the data and compare them to quantified labelling efficiency.

We thank the reviewer for highlighting the issue of whether the probe is sufficiently sensitive to reveal changes in activity that are uncovered by conventional E3 activity assays. We have now optimised probe labelling of RNF12 mutants using a more efficient 365 nm LED lamp, and provide new data demonstrating detectable but reduced probe labelling of RNF12 R599C and R387C (Figure 4C). We also provide a quantitative

comparison of RNF12 lysine discharge activity (Figure 4B) with photoABP labelling efficiency (Figure 4B vs. 4C). These data confirm that results obtained from probe labelling and lysine discharge approaches are quantitatively similar, and consistent with previous lysine discharge data from Bustos et al, Cell Reports (2018). Therefore, we argue that the probe is sufficiently sensitive to detect changes in activity when compared to conventional ubiquitylation assays. An additional advantage with the photoABP assay is it can measure RING E3 ubiquitin ligase activity in cell extracts, which cannot be achieved by conventional ubiquitylation assays. In light of comments from Reviewer 1 (please see above), we have also added text highlighting a caveat with both lysine discharge and probe assays which is that they might not always strictly correlate with protein substrate ubiquitylation assays.

3. While this probe is a very interesting and valuable addition to the arsenal of ubiquitin probes available, the authors should not overstate the claims regarding its use. For example, on page 5 the authors state: "providing new mechanistic insights into how TOKAS variants disrupt catalytic activity." In this paper the photoABP is only used to monitor the presence or absence of activity of RNF12, without providing insight into a molecular mechanism. Similarly, the authors suggest that the photoABP can be used to "robustly monitor differences in RNF12 activity" (page 15), but this is not supported by data, as the low labeling efficiency is unlikely to monitor differences in activity robustly. We again thank the reviewer for this suggestion and have removed the phrase "providing new mechanistic insights into how TOKAS variants disrupt catalytic activity". We hope our new data showing that we can detect the low activity of RNF12 with RNF12 R599C and R387C mutants using the LED lamp sufficiently addresses the reviewer's concerns (Figure 4C).

4. The authors claim throughout the manuscript that the ABP has been tested in patient-derived cell lines. This is misleading because hiPSCs are not cells from TOKAS patients, although they are of human origin. By making this claim, the authors are exaggerating the extent to which their ABP has been used in disease models. This is alluded to by calling these cells a 'prototype patient-derived cell line'. In addition, it should be made clear in the manuscript that ABP incubations were performed in cell extracts; the repeated use of 'in ... cell-lines' suggests that incubations were performed in intact cells. We thank the reviewer for this comment. We clarify that the probe has been used in hiPSCs, which are cells of human origin, rather than TOKAS patient/prototype cell lines, which are not yet available (see abstract). We have also clarified that probe measurements are made *in situ* within cell extracts.

5. In Figure 4G there is a band visible in the WT non-UV lane without the probe at the same position as the probe labelling. Could the authors comment on the presence of non-specific labeling? Unfortunately, in this experiment we find a variable, non-specific band which cannot be attributed to non-specific probe binding as it is present even in the absence of probe and is thus cross reactivity of the antibody. We do not have an explanation for why the non-specific signal is more pronounced in lane 1 and we do not observe this in replicate experiments. We have replaced these data with a repeat of this experiment where the non-specific signal is more consistent:

6. In Figures 5B and 5C, why is RNF12 detected in the NiNTA pull-down samples? The pull-down does not seem to enrich photoABP-RNF12, compared to assays without this step (Figure 4G).

We thank the reviewer for raising this point. As a result of the low RNF12 expression level in hiPSCs (Figure 5A) and high background generated from these cell extracts, this strategy was required to specifically detect the photoABP-RNF12 signal. However, there is a non-specific interaction between RNF12 and the resin, which explains why there is unlabelled RNF12 in the NiNTA pull-down. We now clarify this in the legend for Figure 5B and 5C (p32 line 12).

The authors should describe in the figure legend how this experiment was carried out.

We have clarified the rationale and provide further explanation of the methodology in the figure legend.

There also seems to be a band indicating labeling of the W576Y mutant.

R575C and W576Y are not refractory to probe labelling as can be seen with the new Figure 4G data above. However, probe labelling is consistently reduced for these two mutants relative to WT.

7. In some experiments, the signal for the photoABP in cell lysate assays appears stronger in the RNF12 wild-type sample compared to mutants (Figure 3C; Figure 4C). Did authors use more photoABP in these assays?

We thank the reviewer for pointing this out and can confirm that the amount of probe used is constant between samples. Indeed, we clarify in the figure legends that one reaction mix per condition (\pm UV) is divided in two to ensure that identical amounts of probe are added in each case. We suspect the impression of increased loading is due to broadening of the signal because of the appearance of a lower band in the presence of WT RNF12 and UV. An explanation for this might be the closed conformation stabilized by WT E3 is susceptible to intramolecular crosslinking, which competes with E3 labelling, and the internal crosslink alters the electrophoretic mobility of the probe. We have added a note to the legend proposing this.

Reviewer #3 (Comments to the Authors (Required)):

In their manuscript, Bustos et al. tested the hypothesis if RNF12 RING E3 ubiquitin ligase activity can be assessed by photocrosslinking Activity-Based Probes (photoABPs). They used this assay to investigate if photoABP binding to RNF12 differs in RNF12 mutants that are associated with the developmental disorder Tonne-Kalscheuer syndrome (TOKAS) and found this to be reduced. As a proof of principle this method was used in patient derived cell-lines, however, not from TOKAS patients. The manuscript is well prepared and the experiments are overall well controlled. However, I do have some points that should be addressed before the manuscript is ready for publication.

Major comments:

The E2's UBCH5A [1, 2] and UBC8 [3] are known to cooperate with RNF12. Please explain why UBE2D3 (S22R C85K) and not UBCH5A or UBC8 has been used in the assay. The paper where UBE2D3 was shown to cooperate with RNF12 for substrate degradation needs to be cited. Otherwise it needs to be stated that the assay relies on an unspecific E3 RING - E2 interaction.

We thank the reviewer for raising this important point. We previously showed in a screen of E2s that RNF12 specifically catalyses ubiquitylation in conjunction with the UBE2D/E family (which includes UBE2D3 and UBE2D1/UBCH5A) and UBE2W (Bustos et al, Cell Reports 2018; see data below). As the RING E3 photoABP is an engineered form of UBE2D3, we reasoned that the probe monitors a specific E2-RNF12 RING E3 interaction. However, as the UBE2D/E family of E2s are productive for chain elongation with most RING E3s, it remains possible that this represents a non-specific E2-E3 interaction. We have now clarified this in the text and cited our study to pinpoint RNF12 E2 specificity.

Several targets of RNF12, such as REX1[1], HDAC2 [3], TRF1 [2], Stathmin [4], ERalpha [5], BRF1 [6] and SMAD7 [7] were described in the literature. At least some of the known RNF12 substrates should be mentioned in the manuscript and a reason why REX1 was chosen as RNF12 substrate needs to be provided.

We thank the reviewer for pointing this out. We have now stated that RNF12 ubiquitylates other substrates and referenced the papers. We also clarify that REX1 is likely an RNF12 substrate that is relevant for TOKAS as it controls neurodevelopmental gene expression (Bustos et al, Dev Cell 2020).

For a broader readership background information for RNF12 is required. Please describe

whether or not RNF12 is expressed in a tissue specific manner, if its expression or content is changed during development and in which cellular compartment it is contained. We thank the reviewer for raising this important point. We have now stated that RNF12 is a predominantly nuclear protein with tissue-specific expression including enrichment in the brain.

On page 13 the following is stated "Furthermore, RNF12 RING domain mutations disrupt RNF12-dependent gene expression, as measured by USP26 induction." However, USP26 protein is determined in the assays. In order to state that RNF12-dependent gene expression is changed USP26 expression needs to be measured and reported. We have shown previously that *Usp26* gene expression is regulated by the RNF12-REX1 axis and that USP26 protein levels reflect *Usp26* mRNA levels (Segarra-Fas et al, bioRxiv 2020). We have now clarified in the text.

The title should be focused on the content of the manuscript; namely the usage of photoABPs to analyze RNF12 TOKAS variants. We have modified the title accordingly:

"ACTIVITY-BASED PROBE PROFILING OF RNF12 RING E3 UBIQUITIN LIGASE FUNCTION IN TONNE-KALSCHEUER SYNDROME"

Minor comments:

For the general audience the purpose of the "single turnover lysine E2 ubiquitin discharge assay" needs to be explained.

We have now explained this in more detail in the text.

Page 4: Figure 1A does not show what the authors refer to.

We thank the reviewer for pointing out this error, which we have now corrected.

Figures 3 and 4: Please explain why ERK1 immunoblots were performed.

These are included as loading controls, which we have clarified in the figure legend.

Figure 3 and 4: please indicate what "IR scan" stands for.

We clarify that this stands for "Infra-red scan" to detect IR-800 labelled.

Figure 4F: please show immunoblot for REX1

We have shown previously REX1 accumulation in RNF12 R575C (Bustos et al, Cell Reports 2018; Segarra-Fas et al, bioRxiv 2020) and W576Y (Segarra-Fas et al, bioRxiv 2020) knock-in mESCs. We have now highlighted this in the text and provide the data below.

Figure 4G: please explain the signal in the first lane in IB: RNF12, which migrates at the heights of photoABP-RNF12, and leave questions about the specificity of the assay.

Unfortunately, in this experiment we find a variable, non-specific band which cannot be attributed to non-specific probe binding as it is present even in the absence of probe and is thus cross reactivity of the antibody. We do not have an explanation for why the non-specific signal is a more pronounced in lane 1 and we do not observe this in replicate experiments. We have replaced these data with a repeat of this experiment where the non-specific signal is more consistent (revised figure panel also presented within response to Reviewer 2).

May 3, 2022

Re: Life Science Alliance manuscript #LSA-2021-01248R

Dr. Greg Findlay
MRC Protein Phosphorylation and Ubiquitylation Unit
MRC-PPU
School of Life Sciences
Dundee DD1 5EH
United Kingdom

Dear Dr. Findlay,

Thank you for submitting your revised manuscript entitled "ACTIVITY-BASED PROBE PROFILING OF RNF12 E3 UBIQUITIN LIGASE FUNCTION IN TONNE-KALSCHUEER SYNDROME" to Life Science Alliance. The manuscript has been seen by the original reviewers whose comments are appended below. While the reviewers continue to be overall positive about the work in terms of its suitability for Life Science Alliance, some important issues remain.

Our general policy is that papers are considered through only one revision cycle; however, given that the suggested changes are relatively minor, we are open to one additional short round of revision. Please note that I will expect to make a final decision without additional reviewer input upon re-submission.

Please submit the final revision within one month, along with a letter that includes a point by point response to the remaining reviewer comments.

To upload the revised version of your manuscript, please log in to your account: <https://lsa.msubmit.net/cgi-bin/main.plex>
You will be guided to complete the submission of your revised manuscript and to fill in all necessary information.

B. MANUSCRIPT ORGANIZATION AND FORMATTING:

Sincerely,

Reviewer #1 (Comments to the Authors (Required)):

In their revised manuscript, the authors have clarified several important points and included revised experiments that demonstrate more granular measurements of RNF12 activity using the photoABP. Several of their explanations for strange results, however, were not satisfactory and should be further addressed. In addition, some aspects of the manuscript are still overstated and should be toned down to better reflect the study.

Major Concerns:

The authors propose that the relatively weak photoABP labeling of recombinant RNF12 could be due to a low specific activity of their prepared protein. This is an important point worth testing, as it has large implications for understanding exactly what the probe reactivity represents. One simple way to test this would be to perform the photoABP labeling reaction, then use this material for a standard in vitro ubiquitination assay for comparison alongside a control that did not receive UV treatment. If the authors are correct, then the bulk of the detectable RNF12 ligase activity should be inhibited by the crosslinked photoABP.

In their explanation of the UV-independent ABP-RNF12 band present in Fig. 5C, the authors propose that it could result from either nonspecific antibody staining or some UV-independent ABP reactivity. This is an important distinction, as the presence of UV-independent reactivity has implications for how the ABP can be utilized in the future. The authors should attempt to clarify what this band is. For instance, could DUB treatment of the sample release the E2 and cause a mass shift in this cross-reactive band?

A statement in the Discussion section, "This represents the first use of ABP technology to monitor dysregulated RING E3 activity in human diseases", is overstated and needs to be toned down. Unless you are using the ABP to test activity of endogenous RNF12 from patient samples, I would argue that you have only performed proof-of-principle experiments.

Reviewer #2 (Comments to the Authors (Required)):

The authors have addressed all my concerns and the paper should now be published.
This is a very interesting extension to previous work by the authors that should attract a lot of attention.

Reviewer #3 (Comments to the Authors (Required)):

The authors have sufficiently addressed all comments I had. I do not have any further comments and congratulate the authors on their work.

Reviewer #1 (Comments to the Authors (Required)):

In their revised manuscript, the authors have clarified several important points and included revised experiments that demonstrate more granular measurements of RNF12 activity using the photoABP. Several of their explanations for strange results, however, were not satisfactory and should be further addressed. In addition, some aspects of the manuscript are still overstated and should be toned down to better reflect the study.

Major Concerns:

The authors propose that the relatively weak photoABP labeling of recombinant RNF12 could be due to a low specific activity of their prepared protein. This is an important point worth testing, as it has large implications for understanding exactly what the probe reactivity represents. One simple way to test this would be to perform the photoABP labeling reaction, then use this material for a standard in vitro ubiquitination assay for comparison alongside a control that did not receive UV treatment. If the authors are correct, then the bulk of the detectable RNF12 ligase activity should be inhibited by the crosslinked photoABP.

We thank the reviewer for this insightful suggestion to resolve this issue. We have now compared RNF12 activity \pm UV-induced probe labelling and find that there is no stabilization of E2~Ub after probe labelling, which would be consistent with probe labelling inhibiting RNF12 activity (see data below). This shows that our claim that the low probe labelling is due to a small fraction of RNF12 being active is incorrect. We have revised the text in the attached manuscript and included a supporting reference;

“It should be noted that a minor fraction of RNF12 is probe-labelled in these experiments. This might be on account of the Bpa photocrosslinker’s preference for certain amino acids, propensity to form intramolecular rearrangement products, and that its crosslinking efficiency can be influenced by geometric constraints”

And;

“Again, the minor fraction of RNF12 that is probe-labelled is likely to be due to factors that compromise the efficiency of the crosslinking reaction, rather than being reflective of the absolute stoichiometry of active RNF12.

In their explanation of the UV-independent ABP-RNF12 band present in Fig. 5C, the authors propose that it could result from either nonspecific antibody staining or some UV-independent ABP reactivity. This is an important distinction, as the presence of UV-independent reactivity has implications for

how the ABP can be utilized in the future. The authors should attempt to clarify what this band is. For instance, could DUB treatment of the sample release the E2 and cause a mass shift in this cross-reactive band?

The band identified by the reviewer is extremely weak (see annotated Fig 5C below), suggesting that the potential for non-specific UV independent labelling is minimal. Consistent with this, other experimental replicates do not detect this band (see data provided below). Therefore, this band is so close to background that it is unlikely to represent a major source of UV-independent reactivity.

Current Figure 5C (position of possible UV-independent probe labelling highlighted in red):

Replicate of Figure 5C (no UV-independent probe labelling observed):

A statement in the Discussion section, "This represents the first use of ABP technology to monitor dysregulated RING E3 activity in human diseases", is overstated and needs to be toned down. Unless you are using the ABP to test activity of endogenous RNF12 from patient samples, I would argue that you have only performed proof-of-principle experiments.

We have toned this down by removing or modifying references to patient samples throughout the text. In the specific example highlighted by the reviewer, we have modified the text as follows;

"This provides proof-of-principle for the use of ABP technology to monitor dysregulated RING E3 activity in human diseases."

Reviewer #2 (Comments to the Authors (Required)):

The authors have addressed all my concerns and the paper should now be published. This is a very interesting extension to previous work by the authors that should attract a lot of attention.

Reviewer #3 (Comments to the Authors (Required)):

The authors have sufficiently addressed all comments I had. I do not have any further comments and congratulate the authors on their work. May 3, 2022

June 1, 2022

RE: Life Science Alliance Manuscript #LSA-2021-01248RR

Dr. Greg Findlay
MRC Protein Phosphorylation and Ubiquitylation Unit
MRC-PPU
School of Life Sciences
Dundee DD1 5EH
United Kingdom

Dear Dr. Findlay,

Thank you for submitting your revised manuscript entitled "ACTIVITY-BASED PROBE PROFILING OF RNF12 E3 UBIQUITIN LIGASE FUNCTION IN TONNE-KALSCHEUER SYNDROME". We would be happy to publish your paper in Life Science Alliance pending final revisions necessary to meet our formatting guidelines.

-please consult our manuscript preparation guidelines <https://www.life-science-alliance.org/manuscript-prep> and make sure your manuscript sections are in the correct order

A. FINAL FILES:

B. MANUSCRIPT ORGANIZATION AND FORMATTING:

****It is Life Science Alliance policy that if requested, original data images must be made available to the editors. Failure to provide**

original images upon request will result in unavoidable delays in publication. Please ensure that you have access to all original data images prior to final submission.**

The license to publish form must be signed before your manuscript can be sent to production. A link to the electronic license to publish form will be sent to the corresponding author only. Please take a moment to check your funder requirements.

Sincerely,

June 8, 2022

RE: Life Science Alliance Manuscript #LSA-2021-01248RRR

Dr. Greg Findlay
MRC Protein Phosphorylation and Ubiquitylation Unit
MRC-PPU
School of Life Sciences
Dundee DD1 5EH
United Kingdom

Dear Dr. Findlay,

Thank you for submitting your Research Article entitled "ACTIVITY-BASED PROBE PROFILING OF RNF12 E3 UBIQUITIN LIGASE FUNCTION IN TONNE-KALSCHEUER SYNDROME". It is a pleasure to let you know that your manuscript is now accepted for publication in Life Science Alliance. Congratulations on this interesting work.

DISTRIBUTION OF MATERIALS:

Again, congratulations on a very nice paper. I hope you found the review process to be constructive and are pleased with how the manuscript was handled editorially. We look forward to future exciting submissions from your lab.

Sincerely,
